# Japanese Are Less Human-Centred than French: A New View on Spontaneous Perspective-Taking in Easterners

**DOI:** 10.3390/bs15111482

**Published:** 2025-10-30

**Authors:** François Quesque, Akira Imai, Kenji Susami, Chiharu Niki, Eric Chabanat, Alexandre Foncelle, Jean-Baptiste Van der Henst, Ayumi Kambara, Yves Rossetti

**Affiliations:** 1Inserm UMR-S 1028, CNRS UMR 5292, Trajectoires Team, Centre de Recherche en Neurosciences de Lyon, Université Lyon 1, 16 Avenue Lépine, 69676 Bron, France; francois.quesque@gmail.com (F.Q.); niki.chiharu@twmu.ac.jp (C.N.); eric.chabanat@univ-lyon1.fr (E.C.); jean-baptiste.vanderhenst@cnrs.fr (J.-B.V.d.H.); kanbara.ayumi@kuas.ac.jp (A.K.); 2Plateforme “Mouvement et Handicap” and NeuroImmersion, Hopital Henry-Gabrielle, Hospices Civils de Lyon, 20 route de Vourles, 69002 Saint-Genis-Laval, France; 3Department of Psychology, Faculty of Arts, Shinshu University, Matsumoto 390-8621, Japan; imaiakr@shinshu-u.ac.jp; 4Department of Sociology, Kindai University, Higashiosaka 577-8502, Japan; 5Advanced Telecommunications Research Institute International (ATR), Kyoto 619-0288, Japan; 6Institute of Advanced Biomedical Engineering and Science, Tokyo Women’s Medical University, Tokyo 162-8666, Japan; 7Faculty of Humanities, Kyoto University of Advanced Science, Kyoto 615-8577, Japan

**Keywords:** perspective-taking, cross-cultural differences, collectivism, individualism, egocentrism

## Abstract

Perspective-taking is fundamental to social interaction. In line with psychosocial ideas that Eastern societies value the individual’s interdependence, recent experimental work suggests that they are more inclined to endorse another person’s perspective than people from Western countries. There are, however, more cultural differences between those societies than interdependence. Because Eastern societies also sustain a more holistic nature of cognition, people from Eastern countries may simply tend to interpret the surrounding world from a less ego-centred perspective. Direct support for this idea was found when comparing the responses of Japanese and French participants in level-2 visuo-spatial perspective-taking tasks. As predicted, we observed a less egocentric bias in Japanese than in French participants. Crucially, this bias was not caused directly by a greater proportion endorsing the point of view of another person but rather indirectly by a higher disposition to spontaneously adopt non-human-centred perspectives.

## 1. Introduction

Our physical world has no point of view, but perception is inherently bound to a vantage point. One fundamental question in social psychology is about how individuals may be able to depart from this intrinsic first-person perspective to adopt another perspective. This question encompasses spatial, emotional, and cognitive perspective-taking ([30]), and some empirical data shows that shifting perspective cost also applies to theory-of-mind-related tasks and to both Easterners and Westerners ([2]). Our physical body provides access to numerous physical stimulations thorough sense organs, but each perceptual system is endowed with intrinsic constraints and limitations, among which is the reference frames used to encode sensory information achieved through multisensory integration ([15]). The embodied nature of perception governs our way of perceiving and interacting with our physical and social environment. In addition to this material constraint, our perceptual system is shaped and evolves through our development and therefore also depends on our perceptual experiences. Physical richness of the environment and social stimulations and interactions both contribute to this individual development. In addition, social interactions stimulate our ability to take others’ perspectives at the spatial, emotional, and cognitive level, which together provide fundamental tools to elaborate our social cognition and interactions. As a matter of fact, culture affects not only how we perceive the world (e.g., [24]; [14]; [17]) but also how we perceive others (e.g., [29]) and others’ representations of the world ([39]). Recurrently, culture plays a significant role in shaping the tendency to adopt others’ points of view, a process frequently referred to as perspective-taking or mentalizing, although specific terminology should be adjusted to account for the various facets of this general ability ([27]). There is substantial cross-cultural psychological and anthropological research showing that cultural norms, values, and social practices influence how readily and in what ways people adopt others’ viewpoints (e.g., [39]; [3]).

One of the most robust findings can be described as the contrast between collectivist cultures and individualist cultures. Collectivist cultures (e.g., East Asia, Latin America, Africa) emphasize interdependence, harmony, and group cohesion (e.g., [6]; [18]). People from these cultures are more likely to be attuned to others’ mental states, engage in context-sensitive social reasoning, or adjust their communication based on others’ needs and viewpoints (e.g., [34]). One representative example is provided by unique cultural behaviours such as “Omoiyari” (feelings of considerable thoughts for others, including empathy, sympathy, and compassion) in Japanese ([35]). By contrast, individualist cultures (e.g., U.S., U.K., Western Europe) emphasize independence and self-expression (e.g., [24]). It is therefore often assumed that people in these cultures tend to focus more on personal goals and internal traits or may be less prone to spontaneously take others’ perspectives unless prompted (e.g., [39]; [19]). These general tendencies are also accompanied with more subtle variations (e.g., [1]; [2]; [20]).

In this context, [25] ([25]) showed that Japanese children outperformed American children on tasks requiring inference of others’ mental states, suggesting cultural shaping of early social cognition. [16] ([16]) compared visual perspective-taking between Germans and Chinese. They showed that Chinese participants more accurately tracked others’ perspectives, especially in social tasks. This may imply a greater automaticity in perspective-shifting in collectivist cultures. [39] ([39]) showed that Chinese take another person’s perspective more spontaneously than Americans when interpreting ambiguous information. This result has been interpreted as an inclination to represent more interdependent selves among members of collectivistic cultures compared to individualistic cultures ([4]; [5]; [22]). The spontaneous nature of this mechanism should, however, be considered with caution. [38] ([38]) also showed that this capacity does not appear at the level of integration but at a later stage of processing, only after the egocentric coding has been inhibited or overtaken (i.e., correction mechanism).

Because of a greater sense of interdependence, Easterners tend to engage in context-dependent and holistic perceptual processes. In contrast, Westerners tend to engage in context-independent and analytic perceptual processes by focusing on a salient object independently of its context ([24]; [23]; [26]). To adopt the point of view of another person typically primarily requires inhibition of one’s own point of view and the ability to consider alternatives ([8]; See [32] for a discussion). Therefore, one may wonder whether allocating attention to the context may simply enable Easterners to adopt a non-ego-centered perspective or more specifically facilitate the adoption of someone else’s perspective. In the experiment mentioned above, [39] ([39]) focused on the visibility of items, which corresponds to “level-1” visuo-spatial perspective-taking (i.e., representing if items are visible or not for others; ([10]; see also [27]) for a consensual lexicon regarding mental states inference processes).

If one wants to interpret data about spatial perspective-taking as a tendency to adopt someone else’s perspective, one has to carefully disentangle between this ability and the mere ability to adopt a non-ego-centred perspective. As a matter of fact, object-centred or geo-centred reference frames may also be used and give rise to misleading interpretations. Hence, adopting a decentred perspective does not necessarily imply that another person’s perspective is considered. One strategy is therefore to focus on “level-2” visuo-spatial perspective-taking (i.e., representing how others see the world: “This object is on your left and close to you even if I see it on my right and far from me”) ([11]; [27]). Along this line, we adapted a validated implicit task for the investigation of spontaneous visuo-spatial perspective-taking that is known to elicit a small but significant proportion of alter-centred responses in Westerners ([12]; [28], [30]; [36]; [37]; [40]), i.e., responses for which the spatial scene is considered from the other person’s vantage point. Our participants were confronted with a visual scene containing a person and were asked to localize or to identify a target object with respect to another landmark object in the scene (Bottle task, see [37]; [28], [30]). As a control task, we used an ambiguous number task (6 vs. 9) presented in a social scene ([28]), for which only egocentric and allocentric responses can be obtained. The basic idea was that the bottle task (where is the bottle?) should give rise to a broader variety of responses than simply egocentric vs. allocentric (left vs. right). Therefore, the bottle task may reveal additional information that cannot be obtained from the number task. Therefore, we investigated here whether Easterners exhibit a decreased tendency to consider an ego-centred perspective or a veridical and specific inclination to take another human’s perspective.

## 2. Method

### 2.1. Participants

One hundred and eighty-two French psychology undergraduate students from the university of Lille (mean age = 20.8, SD =2.7, 127 women) and one hundred and eighty-eight Japanese psychology undergraduate students from the university of Shinshu (Matsumoto) and Kinki (Kyoto) were recruited as participants (mean age = 19.3, SD =1.6, 98 women). This study was presented as aiming to collect spontaneous descriptions of diverse visual scenes in order to develop a standardized verbal fluency battery. Participants had no prior knowledge about the scientific aim of the study. Students volunteered to participate at the end of their lecture. They were given paper questionnaires including informed consent and questionnaires, and then presented with visual stimuli with a video projector. We controlled for the effect of participants’ gender and its interaction with participants’ origin for the two tasks and observed no significant differences (all Z < 1). Sample size was based on earlier studies testing spatial perspective-taking and relying on the same experimental paradigm ([12]; [37]). In addition, a sensitivity power analysis (alpha significance criterion: 0.05, two-tailed; standard power criterion: 80%) computed for the present sample (*n* = 370) allows us to detect Cramer’s V effect size higher than 0.17 (G-power; [9]). All measures, manipulations, and exclusions are reported in the present report. The protocols were in accordance with the principles of the Declaration of Helsinki.

### 2.2. Procedure and Materials

Participants were sequentially presented with two projected scenes. The Digit Scene served as a control to replicate the stronger ego-centric bias among Westerners. It was presented after the Bottle Scene so as not to diminish participants’ tendency to adopt an allo-centred perspective. The bottle scene (A, The Bottle Scene) depicted a table on which were displayed a book and a bottle, with a man reaching for the book ([7]; [37]; [28], [30]). The second scene (B, The Digit Scene) depicted an array of 15 customs cards displayed on a table, with a woman looking at the central card (see Figure 1). In the Digit Scene, the central card displayed a symbol that could be interpreted either as a “6” or as a “9” depending on the adopted perspective (i.e., the participant’s or the actor’s perspective). To control for potential group or ethnic bias, both scenes could involve either French or Japanese actors. Participants were randomly presented with stimuli involving either individuals from their own ethnic group (e.g., Japanese for Japanese) or individuals from a different ethnic group (e.g., French for Japanese) resulting in a 2 × 2 between-subject design. For the Bottle Scene, participants were asked ‘‘*In relation to the bottle, where is the book*?”, and for the Digit Scene they were asked “*Among the symbols that are displayed on the cards, there is a number. What is this number?*”. Participants were free to answer with as many words as they wanted and were not under time pressure.

## 3. Results

Responses were scored as “self perspective” when they were unambiguously given from the participant’s viewpoint (e.g., “to the right” for the Bottle Scene, “6” for the Digit Scene) and as “other perspective” when they were given from the actor’s viewpoint (e.g., “to the left” for the Bottle Scene, “9” for the Digit Scene), and as neutral when they did not refer to any of the human perspectives listed above (e.g., “nearer to the edge of the table”, “at the same level”, etc). We initially intended to consider double perspective responses, which would express the ability to consciously take the perspective of the actor in the picture. However, the total number of double perspective responses was similar in the two groups and for the two tasks. We therefore followed [37] ([37])’s procedure and took their first answer into consideration. Answers referring to both the participant’s and the actor’s perspectives (12.1%) were recoded depending on the first of the two perspectives mentioned, following the approach adopted by [37]’s ([37]) seminal paper (e.g., a self-perspective expressed first was recoded as self-perspective). We were not able to interpret the perspective endorsed by 6 participants for the Bottle Scene and 20 participants for the Digit Scene because their answers were irrelevant (e.g., “The book” for the Bottle Scene, “The number is on the card in the middle” for the Digit Scene) and as a consequence their data was discarded from analysis. In addition, 15 Japanese participants did not respond to the question related to the Bottle Scene. A total number of 364 participants for the Bottle Scene and of 335 participants for the Digit Scenes were included in the analyses. Scored response proportions (Self, Other, and Neutral) were then compared across the conditions (same group actor, different group actor) and the origin of the participants (French, Japanese) with a Pearson’s Chi-square test.

When a significant difference was observed between flat theoretical distributions and observed distributions, two by two comparisons were conducted using the following formula:$\;Z=\frac{P1-P2}{\surd (P\;*\left(1-P\right)\;*\left(\frac{1}{n1}+\frac{1}{n2}\right))}$, with P1 and P2 the respective probabilities to respond with the ego-centered perspective in the two compared conditions, P the mean probability to respond with that perspective, and n1 and n2 the respective total sample size of the compared conditions ([31]). Effect sizes were computed using Cramer’s V for the Chi-square tests. For multiple comparisons, the significance criterion was corrected (*α* < 0.05/n, with n being the number of comparisons).

*“Are Japanese participants less ego-centred than French?” (Digit Scene responses)*. As can be seen in Figure 2, we observed a highly significant deviation of the responses from the flat distribution (χ^2^(2) = 187.6, *p* < 0.001, V = 0.75). Overall, Japanese participants used a significantly less ego-centred perspective than French (Z = 2.67, *p* < 0.01). We also examined for group bias with respect to the actor in the photograph. For Japanese participants, no significant difference was observed between Scenes containing a French or a Japanese person (Z = 0.19, n.s.). Conversely, French participants tended to use significantly more their own perspective when describing a visual scene that contains a French person than a Japanese (Z = 2.4, *p* < 0.01).

*“Are Japanese participants less human-centred than French?” (Bottle Scene responses)*. As it can be seen in Figure 3, we observed a highly significant deviation of the responses from the flat distribution (χ^2^(2) = 273.6, *p* < 0.001, V = 0.87). Again, Japanese participants used a significantly less ego-centred perspective than French (Z = 5.13, *p* < 0.001). The tendency of French participants to use their own perspective did not significantly differ depending on the origin of the actor (Z = 1.23, n.s.) nor did it for Japanese participants (Z = 0.85, n.s.). More importantly, the lower tendency of Japanese participants to use an ego-centred perspective when describing the visual scene does not depend on a stronger tendency to endorse the actor’s perspective (Z = 0.19, n.s.) but results from a greater inclination than French to use an allo-centred perspective (neutral responses, Z = 6.63, *p* < 0.001). Moreover, this inclination was greater when confronted to a French than to a Japanese actor (Z = 3.18, *p* < 0.001).

## 4. Discussion

Our goal was to investigate whether the increased ability of Easterners (relative to Westerners) to adopt another perspective reflects a disposition to adopt the perspective of other humans or results indirectly from a general tendency to engage in more holistic perceptual processes. Whereas previous studies tested whether Easterners were better than Westerners at representing *if* a confederate can or cannot see target objects (e.g., [39]), here we expand this finding by showing that they were more inclined to represent *how* this confederate would perceive the world (see also ([16]) for congruent findings on response time). This less ego-centred tendency of Japanese to represent the world was observed for our two scenes. In addition, the Bottle Scene allowed us to critically observe an important proportion of allo-centred responses, specifically in our Japanese group. The lower tendency of Japanese to adopt an ego-centred perspective then does not depend on a stronger tendency to endorse the actor’s perspective but to a general disposition for allo-centred perspectives. This observation is congruent with the experimental confirmation that Easterners process visual scenes in a more holistic manner than Westerners (e.g., [14]). Contrasting with this, the French proportion of allo-centred responses was minimal and comparable to other Western cultures ([12]; [37]). Overall, this suggests that Easterners are not only less ego-centred but also less human-centred than Westerners, i.e., that they may rely more on physical or geographical landmarks in the environment. This conclusion is in line with religious and philosophical ideas of a continuum between nature and humans (e.g., [24]). Interestingly, French participants more often took the actor’s perspective when facing Japanese but experimental parameters did not modulate their number of allo-centred responses. By contrast, it was the number of allo-centred responses that increased when Japanese were confronted with a French actor. Although our study identified this particular pattern, the origins of such modulation remain highly speculative as they could reflect genuine psychological differences, such as cultural traits, as well as more superficial effects, such as mere differences in the levels of familiarity with other ethnicities (e.g., [16]). This result should be confirmed and extended in order to identify its potential bases. Related to this issue, one may wonder whether the transparency bias, i.e., the illusion that people closer to oneself will better understand their emotions, thoughts or messages ([13]), would be stronger in Western than in Eastern cultures. We reveal the cultural dependency of this previously observed trend ([33]; [36]). Social distance may thus increase the inclination to be less ego-centred in Westerners, whereas it may encourage Easterners to adopt less human-centred perspectives. This last point even questions the nature of spontaneous perspective-taking that could be conceived broadly as a decentred perception of the world, which is facilitated in Eastern cultures. Taken together, the tendency of taking a less ego-centred perspective may also provide a basis for the unique mindfulness of individuals from Buddhist culture ([21]), in which the place of humans in nature is not conceived of in a hierarchical fashion ([35]; [24]). In contrast to individualist societies, collectivist, and specifically Taoist or Buddhist cultures, may promote both allo-centric (i.e., not centred on another human) and non-human centred perspectives. At a broader scale, our results also confirm the high cultural variability of tests used to assess mentalizing and perspective-taking ([29]), which has remained strikingly and severely neglected thus far.

## Figures and Tables

**Figure 1 behavsci-15-01482-f001:**
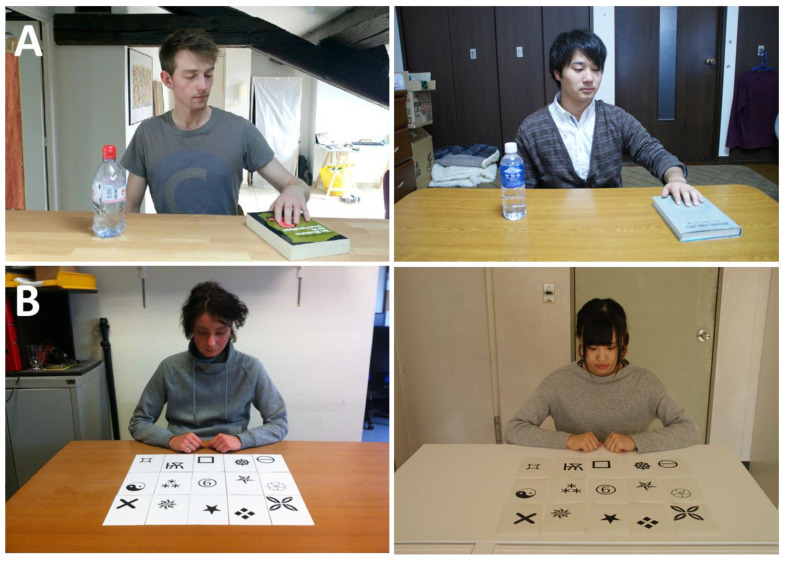
Scenes used depicting (**A**) a male actor reaching for a book displayed on a table on which there is also a bottle ([37]; [28], [30]), (**B**) a female actor heading toward symbols displayed on a table ([28]). Each scene could involve a French actor (**left** panel) or a Japanese actor (**right** panel).

**Figure 2 behavsci-15-01482-f002:**
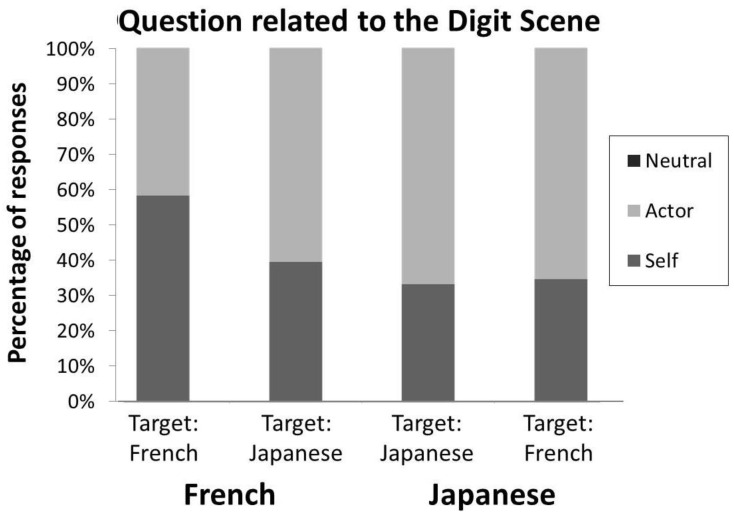
Proportion of responses from an other (Actor), an ego, (Self) or a allo-centred (Neutral) perspective as a function of the origin of participants and of the origin of the image actor (see [28]). French participants produced more frequent ego-centred responses, especially for French actors. Note that no Neutral response was obtained in this condition.

**Figure 3 behavsci-15-01482-f003:**
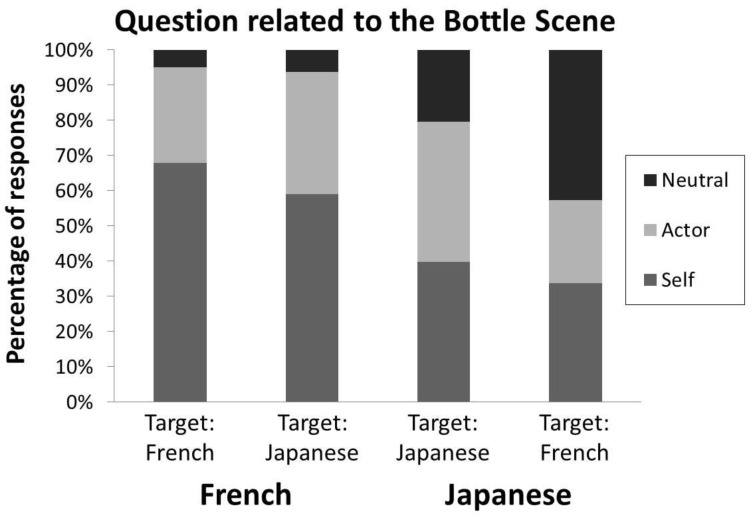
Proportion of responses from an other (Actor)-, ego (Self)-, or allo-centred (Neutral) perspective as a function of the origin of participants and of the origin of the image actor (see [37]; [30]). French participants exhibited a majority of ego-centred responses, whereas the Japanese participants produced more neutral responses than the French.

## Data Availability

The original contributions presented in this study are included in the article. Further inquiries can be directed to the corresponding author.

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
