# Peer review of "Japanese Are Less Human-Centred than French: A New View on Spontaneous Perspective-Taking in Easterners"

_behavsci, 2025, doi:10.3390/bs15111482_

Round 1

Reviewer 1 Report

Comments and Suggestions for Authors

The study compared Western and Eastern visuospatial perspective-taking abilities and concluded that individuals from the Eastern country exhibited fewer egocentric biases. The study appears methodologically sound and presents interesting findings. However, the major weakness of the manuscript lies in the theoretical section. It lacks a clear theoretical line of argument, explanatory depth, and a clear description of the study’s aims. Furthermore, the manuscript does not present clear hypotheses and might omit relevant citations on how culture influences perspective-taking. The core idea is promising, but several theoretical, methodological, and interpretational aspects require clarification.

Introduction

  • The theory section is very short. I would recommend expanding it to more thoroughly lay out your theoretical framework in more detail. The introduction currently focuses primarily on the research question and the means to test it, rather than explaining why the expected effect should occur and what mechanisms might underlie them.
  • Please be more specific on the type of perspective-taking you are addressing (define viso-spatial perspective-taking and differentiate it from other forms (e.g., cognitive, affective) of perspective-taking)
  • A mere suggestion: Maybe start with a more easy-to-follow example in the beginning to improve readability; the current opening is very technical
  • 2 lines 48-54: Please elaborate how the tasks test which prediction precisely. The first sentence is not a hypothesis but a research question; consider reformulating it accordingly.
  • 2. lines 55-58: Please clarify how the task elicits “allo-centred” responses. Be more specific about the underlying theoretical rationale. Also, maybe define allo-centred.

After reading the full manuscript:

  • Please include hypotheses explicitly as you test them in the results section
  • I would strongly suggest addressing (and explaining!) the potential ethnicity bias in the theory section, as this is a crucial novelty of your study

Method

  • Please refer to participants as “women” instead of “females.”
  • Power: please state the sample size N for the sensitivity analysis. I stumbled over “meaningfully report” (p. 2, line 75), consider rephrasing it to something like “effect that can be detected”. In addition, maybe state the range of the effect sizes of the studies you based your sample on.
  • Please specify: Was it a lab or an online study, was is computer-based?
  • Please specify: Which software was used?
  • Please specify: How did you recruit your participants?
  • Please specify: How were the participants compensated?
  • Please specify: Did you preregister the study?
  • Please provide a clearer description of your design (2x2, which are the IVs and the factor levels for each variable);à 4 line 120, While the conditions are described more clearly here, please include a detailed description of the design in the method section
  • p. 3, lines 94-98: “The Digit Scene served…”: These sentences seem to be misplaced, I recommend moving them up to the presentation of the present research (end of introduction), it would also help to argue in more detail why the one task tests ego-centrism, and the other human-centrism and how this might depend on ethnicity!

Results:

  • Coding: I would be interested if it made a difference for your results whether or not you include the data of participants who referred to both perspectives. These participants likely show reduced bias, and their inclusion might alter observed effects.
  • Rather than structuring the analysis around research questions, I recommend formulating explicit hypotheses in the introduction and referring back to them here. This would enhance theoretical coherence and facilitate interpretation.
  • Please refer to figures 2 and 3 explicitly in the text
  • 7, 183-186: This is an interesting finding. I would suggest supporting your potential explanation with references and consider proposing a follow-up study to further explore this mechanism.
  • 7., 187 ff: (Taken together): The use of Omoiyari as an explanatory example seems misplaced. Given that your data suggest a reduction in egocentric bias due to increased neutral or allo-centred responses, rather than a stronger other-centred tendency, this example may not fit the empirical pattern.

Minor:

  • Few minor grammatical errors/typos (e.g., page 6, Line 166 “if [a] confederate”), please do a thorough spell/grammar check at the end

I thank the authors for their work and the opportunity to read and evaluate this manuscript.

Author Response

Reviewer 1

The study compared Western and Eastern visuospatial perspective-taking abilities and concluded that individuals from the Eastern country exhibited fewer egocentric biases. The study appears methodologically sound and presents interesting findings. However, the major weakness of the manuscript lies in the theoretical section. It lacks a clear theoretical line of argument, explanatory depth, and a clear description of the study’s aims. Furthermore, the manuscript does not present clear hypotheses and might omit relevant citations on how culture influences perspective-taking. The core idea is promising, but several theoretical, methodological, and interpretational aspects require clarification.

Response: We initially aimed at being as concise as possible, and we wish to thank the reviewer for identifying which crucial information was missing from our initial ms.

Introduction

  • The theory section is very short. I would recommend expanding it to more thoroughly lay out your theoretical framework in more detail. The introduction currently focuses primarily on the research question and the means to test it, rather than explaining why the expected effect should occur and what mechanisms might underlie them.
  • Please be more specific on the type of perspective-taking you are addressing (define viso-spatial perspective-taking and differentiate it from other forms (e.g., cognitive, affective) of perspective-taking)
  • A mere suggestion: Maybe start with a more easy-to-follow example in the beginning to improve readability; the current opening is very technical
  • 2 lines 48-54: Please elaborate how the tasks test which prediction precisely. The first sentence is not a hypothesis but a research question; consider reformulating it accordingly.
  • 2. lines 55-58: Please clarify how the task elicits “allo-centred” responses. Be more specific about the underlying theoretical rationale. Also, maybe define allo-centred.

Response: Thank you for pointing out this improvement for our introduction section. We initially aimed at being as concise as possible, but our revised introduction addresses your points and is now providing useful information to the reader.

After reading the full manuscript:

  • Please include hypotheses explicitly as you test them in the results section

Response: Subtitles in the result section now correspond to the hypotheses.

  • I would strongly suggest addressing (and explaining!) the potential ethnicity bias in the theory section, as this is a crucial novelty of your study

Response: We have now provided a short discussion of this finding.

Method

  • Please refer to participants as “women” instead of “females.”

Response: done.

  • Please specify: Which software was used?

Response: added.

  • Please specify: How did you recruit your participants? How were the participants compensated?

Response: Details provided.

  • Please specify: Was it a lab or an online study, was is computer-based?

Response: explained.

  • Please provide a clearer description of your design (2x2, which are the IVs and the factor levels for each variable);à 4 line 120, While the conditions are described more clearly here, please include a detailed description of the design in the method section

Response: our description has been enhanced.

  • Please specify: Did you preregister the study?

Response: corresponding details are now included in the revision.

  • Power: please state the sample size N for the sensitivity analysis. I stumbled over “meaningfully report” (p. 2, line 75), consider rephrasing it to something like “effect that can be detected”. In addition, maybe state the range of the effect sizes of the studies you based your sample on.

Response: We revised the section accordingly.

  • p. 3, lines 94-98: “The Digit Scene served…”: These sentences seem to be misplaced, I recommend moving them up to the presentation of the present research (end of introduction), it would also help to argue in more detail why the one task tests ego-centrism, and the other human-centrism and how this might depend on ethnicity!

Response: Thank you for this nice suggestion.

Results:

  • Coding: I would be interested if it made a difference for your results whether or not you include the data of participants who referred to both perspectives. These participants likely show reduced bias, and their inclusion might alter observed effects.

Response: we also intended to consider this dimension of the responses but there was no significant effect of group on the double perspective responses.

  • Rather than structuring the analysis around research questions, I recommend formulating explicit hypotheses in the introduction and referring back to them here. This would enhance theoretical coherence and facilitate interpretation.

Response: Thank you for this useful recommendation.

  • Please refer to figures 2 and 3 explicitly in the text

Response: Thank you for pointing this beginners’ mistake out! This is now the case.

  • 7, 183-186: This is an interesting finding. I would suggest supporting your potential explanation with references and consider proposing a follow-up study to further explore this mechanism.

Response: Thank your liking this observation. We have now added details on this point.

  • 7., 187 ff: (Taken together): The use of Omoiyari as an explanatory example seems misplaced. Given that your data suggest a reduction in egocentric bias due to increased neutral or allo-centred responses, rather than a stronger other-centred tendency, this example may not fit the empirical pattern.

Response: We have now moved this issue in the introduction. Thank you for your constructive criticism.

Minor:

  • Few minor grammatical errors/typos (e.g., page 6, Line 166 “if [a] confederate”), please do a thorough spell/grammar check at the end

Response: Thank you!

Reviewer 2 Report

Comments and Suggestions for Authors

This manuscript presents an interesting interpretation of cross-cultural differences in perspective-taking between Eastern and Western participants. Rather than attributing Easterners’ lower rates of egocentric responses solely to collectivist, other-centered tendencies, the authors propose a broader explanation rooted in holistic cognitive style. This theoretical direction is compelling, and the experimental design has the potential to support it. While the study is promising, I have some concerns regarding the task instructions and their potential impact on the subsequent analyses. Additionally, the manuscript would benefit from further improvements in writing. For example, the introduction could provide more background to justify the chosen research design, and several key aspects of the findings could be discussed in greater depth in the General Discussion. Overall, I believe this manuscript has potential for publication following moderate revisions and clarifications.

My main concern lies in the task instruction and its implications for the analysis in the Digit Scene. In this task, participants were asked, "Among the symbols that are displayed on the cards, there is a number. What is this number?"—a closed-ended question that clearly limits the response to either “6” or “9”. In contrast to the Bottle Scene, where the “where” question allows for the possibility of neutral responses (i.e., neither self nor other-oriented, possibly object-based), the task's formulation in the Digit Scene inherently excludes neutral responses. This asymmetry is confirmed in the results, where neutral responses are completely absent from the Digit task. In fact, prior research using similar digit-based tasks has also consistently elicited binary responses (e.g. Zhao et al., 2015 https://escholarship.org/uc/item/7ts6j98x). Therefore, I question the decision to evaluate the distribution of responses in this task using a flat distribution across three categories (self/other/neutral). A more appropriate baseline might be a 50/50 distribution between self- and other-perspective responses. At the very least, it would be helpful for the authors to revisit this analytic approach and discuss the methodological limitations it may introduce.

Additionally, several findings are under-discussed, despite being informative. For instance, while the main results support the authors’ claim that Japanese participants were more likely to provide neutral responses in the Bottle Scene and French participants were more egocentric in the Digit Scene, unexpected patterns remain unaddressed. Why, for example, did French participants show stronger egocentrism when the actor was an in-group member (i.e., French) in the Digit Scene, but not in the Bottle Scene? Why did Japanese participants provide more neutral responses when the actor was from the out-group (i.e., French) rather than the in-group (i.e., Japanese)? Even if these effects are not central to the paper’s primary hypothesis, it would still be valuable for the authors to acknowledge and offer some interpretation of them.

Additional comments/concerns:

1 The introduction is overly brief and lacks key conceptual grounding, which may lead to confusion for readers. For instance, the authors claim that level-2 perspective-taking tasks allow for distinguishing between a “non-ego-centered cognitive style” and a “more other-centered tendency,” yet they do not clearly explain why level-1 tasks are insufficient for making this distinction. One plausible explanation—based on my understanding of the literature—is that level-1 tasks typically involve binary judgments (e.g., whether another person can or cannot see an object), whereas level-2 tasks often require judgments about how an object appears from another’s viewpoint. Depending on the specific paradigm, such tasks can elicit responses from three categories: self-perspective, other-perspective, and a third category such as object-based descriptions (neutral). The paradigm developed by Tversky and Hard (2009) is a good example of this. While this interpretation is my own, it is unclear whether it aligns with the authors’ intended reasoning. This issue is important, as not all level-2 tasks offer this level of granularity—some, like the digit scene is always considered to result in binary outcomes. Therefore, it would be helpful for the authors to clarify not only why a level-2 task was chosen, but also why this particular paradigm was considered appropriate for addressing their research question.

The authors propose that East Asians’ reduced egocentrism stems from a holistic cognitive style rather than a socially driven other-centered tendency, which is a reasonable interpretation. However, they mention holistic cognition only briefly (e.g., Nisbett & Miyamoto, 2005) without clearly linking it to the present task. A more detailed discussion of evidence from non-social domains (e.g., visual search, spatial frame of reference tasks) demonstrating context-sensitivity processing in East Asians would further strengthen their argument.

2.There is some lack of clarity in the Methods section regarding the experimental design. The authors describe it as a 2×2 between-subjects design (Participant’s ethnic group×Actor’s ethnic group), but in fact, Task Type (Bottle vs. Digit) appears to be a within-subjects factor, as all participants completed both tasks and the authors’ hypotheses imply differential performance across them. This suggests that the actual design is a 2 (Participant’s ethnic group)×2 (Actor’s ethnic group)×2 (Task Type) mixed design. Additionally, the authors note that performing the Digit task first may reduce allo-centered responses in the Bottle task, and therefore chose to fix the task order. However, it is unclear why Task Type was not treated as a between-subjects factor instead. Assigning participants to only one of the two tasks would have helped eliminate potential carryover effects. I believe it would be helpful for the authors to discuss this design choice in more detail.

3.Since the Digit task followed the Bottle task and was treated as a control, it would be clearer to report Bottle results first, matching the task order. Also, consider combining both task results into one figure to facilitate comparison. As discussed above, it may also be clearer to remove the “neutral” category from the legend in Figure 1, given that the Digit task yields only binary responses.

4.Lastly, I noticed some inconsistency in the terminology used to describe neither self nor other-centered responses (e.g., “neutral”, “allo-centred”, “non-human centred”). Some of these terms—particularly “non-human centred”—are not commonly used in the field. It may be helpful for the authors to aim for more consistent terminology or to provide brief clarifications to ensure their intended meaning is clear to readers.

That’s all! I find the paper conceptually interesting and hope these suggestions will be helpful as the authors continue to refine their work.

Author Response

Reviewer 2

This manuscript presents an interesting interpretation of cross-cultural differences in perspective-taking between Eastern and Western participants. Rather than attributing Easterners’ lower rates of egocentric responses solely to collectivist, other-centered tendencies, the authors propose a broader explanation rooted in holistic cognitive style. This theoretical direction is compelling, and the experimental design has the potential to support it. While the study is promising, I have some concerns regarding the task instructions and their potential impact on the subsequent analyses. Additionally, the manuscript would benefit from further improvements in writing. For example, the introduction could provide more background to justify the chosen research design, and several key aspects of the findings could be discussed in greater depth in the General Discussion. Overall, I believe this manuscript has potential for publication following moderate revisions and clarifications.

My main concern lies in the task instruction and its implications for the analysis in the Digit Scene. In this task, participants were asked, "Among the symbols that are displayed on the cards, there is a number. What is this number?"—a closed-ended question that clearly limits the response to either “6” or “9”. In contrast to the Bottle Scene, where the “where” question allows for the possibility of neutral responses (i.e., neither self nor other-oriented, possibly object-based), the task's formulation in the Digit Scene inherently excludes neutral responses. This asymmetry is confirmed in the results, where neutral responses are completely absent from the Digit task. In fact, prior research using similar digit-based tasks has also consistently elicited binary responses (e.g. Zhao et al., 2015 https://escholarship.org/uc/item/7ts6j98x). Therefore, I question the decision to evaluate the distribution of responses in this task using a flat distribution across three categories (self/other/neutral). A more appropriate baseline might be a 50/50 distribution between self- and other-perspective responses. At the very least, it would be helpful for the authors to revisit this analytic approach and discuss the methodological limitations it may introduce.

Response: We understand your point. The two tasks do not require a similar type of answer and the bottle task appears to be more open than the digit task. Thus, it is expected that a non-human-centered perspective can be expressed only in the bottle task. In fact, one would need to use a symbol that can be read from more than two vantage points in order to render the digit task more open. However, our intention was not to compare the two tasks, and rather to focus only on the two groups. Therefore, the fact that the two tasks are of different nature does should not be relevant in our study. Also, we wish to thank you for bringing this reference to our knowledge. It is now included in our revised version. Introduction as also been fully rewritten based on both reviewers’ comments.

Additionally, several findings are under-discussed, despite being informative. For instance, while the main results support the authors’ claim that Japanese participants were more likely to provide neutral responses in the Bottle Scene and French participants were more egocentric in the Digit Scene, unexpected patterns remain unaddressed. Why, for example, did French participants show stronger egocentrism when the actor was an in-group member (i.e., French) in the Digit Scene, but not in the Bottle Scene? Why did Japanese participants provide more neutral responses when the actor was from the out-group (i.e., French) rather than the in-group (i.e., Japanese)? Even if these effects are not central to the paper’s primary hypothesis, it would still be valuable for the authors to acknowledge and offer some interpretation of them.

Response: Thank you for pointing these interesting features of our results. We have now addressed this issue with more details in the discussion section.

Additional comments/concerns:

1 The introduction is overly brief and lacks key conceptual grounding, which may lead to confusion for readers. For instance, the authors claim that level-2 perspective-taking tasks allow for distinguishing between a “non-ego-centered cognitive style” and a “more other-centered tendency,” yet they do not clearly explain why level-1 tasks are insufficient for making this distinction. One plausible explanation—based on my understanding of the literature—is that level-1 tasks typically involve binary judgments (e.g., whether another person can or cannot see an object), whereas level-2 tasks often require judgments about how an object appears from another’s viewpoint. Depending on the specific paradigm, such tasks can elicit responses from three categories: self-perspective, other-perspective, and a third category such as object-based descriptions (neutral). The paradigm developed by Tversky and Hard (2009) is a good example of this. While this interpretation is my own, it is unclear whether it aligns with the authors’ intended reasoning. This issue is important, as not all level-2 tasks offer this level of granularity—some, like the digit scene is always considered to result in binary outcomes. Therefore, it would be helpful for the authors to clarify not only why a level-2 task was chosen, but also why this particular paradigm was considered appropriate for addressing their research question.

Response: We may not understand your concern fully:  It does not seem to us that all level-2 tasks should be more open than level-1 tasks. For example, the Michelon and Zacks (2006) task that is frequently used in the field to assess level-2 spatial perspective taking requires participants to choose between left and right. Once again, our intention was not to compare the two tasks but to compare two cultural groups. Therefore, the digit task was only used to ensure that our participants sample followed the classically described trend.

The authors propose that East Asians’ reduced egocentrism stems from a holistic cognitive style rather than a socially driven other-centered tendency, which is a reasonable interpretation. However, they mention holistic cognition only briefly (e.g., Nisbett & Miyamoto, 2005) without clearly linking it to the present task. A more detailed discussion of evidence from non-social domains (e.g., visual search, spatial frame of reference tasks) demonstrating context-sensitivity processing in East Asians would further strengthen their argument.

Response: Thank you for enabling us to add more details and discussion on this interesting point.

2.There is some lack of clarity in the Methods section regarding the experimental design. The authors describe it as a 2×2 between-subjects design (Participant’s ethnic group×Actor’s ethnic group), but in fact, Task Type (Bottle vs. Digit) appears to be a within-subjects factor, as all participants completed both tasks and the authors’ hypotheses imply differential performance across them. This suggests that the actual design is a 2 (Participant’s ethnic group)×2 (Actor’s ethnic group)×2 (Task Type) mixed design. Additionally, the authors note that performing the Digit task first may reduce allo-centered responses in the Bottle task, and therefore chose to fix the task order. However, it is unclear why Task Type was not treated as a between-subjects factor instead. Assigning participants to only one of the two tasks would have helped eliminate potential carryover effects. I believe it would be helpful for the authors to discuss this design choice in more detail.

Response: We confirm that we used a 2×2 between-subjects design as the two tasks are not directly compared given to different modality of responses.  As we detailed above, our intention was not to compare the two tasks but to compare two cultural groups on two tasks. The digit task was only used to ensure that our participants sample followed the classically described trend.

3.Since the Digit task followed the Bottle task and was treated as a control, it would be clearer to report Bottle results first, matching the task order. Also, consider combining both task results into one figure to facilitate comparison. As discussed above, it may also be clearer to remove the “neutral” category from the legend in Figure 1, given that the Digit task yields only binary responses.

Response: This commentary seems to focus again on the binary vs open question difference between our two tasks. Obviously following the reviewer’s suggestion would lead to exclude the most interesting result from our data set. We therefore chose not to follow this suggestion.

4.Lastly, I noticed some inconsistency in the terminology used to describe neither self nor other-centered responses (e.g., “neutral”, “allo-centred”, “non-human centred”). Some of these terms—particularly “non-human centred”—are not commonly used in the field. It may be helpful for the authors to aim for more consistent terminology or to provide brief clarifications to ensure their intended meaning is clear to readers.

Response: We have now better defined these terms.

That’s all! I find the paper conceptually interesting and hope these suggestions will be helpful as the authors continue to refine their work.

Response: Thank you for your constructive comments!

Round 2

Reviewer 1 Report

Comments and Suggestions for Authors

I thank the authors for their thorough reply and the revised manuscript. It profited from the revision. Unfortunately, I think it is still not yet publishable, especially regarding the justification and formulation of the hypothesis.

  • The introductory sentences up until line 44 read vaguely and do not help to understand the topic. I suggest getting to the heart of the research question more quickly
  • Many sentences lack proper referencing, especially:  ending at line 55 (several!), ending 58, 60, 64, 66, 81, 95 (!).
  • revise sentence lines 77-79, it does not integrate and should be more informational. I think it is worth spending a sentence more on explaining the role of spontaneity 
  • Lines 100-109: The combination of "allowed us to demonstrate" and the formulation of a hypothesis does not fit. I am still not happy with the formulation of the hypothesis; it still does not align with the result section. Please revise and be more specific (H1: We expect that ... are less ...compared to..., H2: ...). 
  • Theory summary: It is still not clear WHY you expect what you expect, and you are still too vague in formulating a precise prediction
  • Please be more scientifically neutral in your interpretation of lines 254-256. "Humbleness" reads a bit normatively influenced.

Thank you for the opportunity to evaluate the revision!

Author Response

Responses to the reviewer :

I thank the authors for their thorough reply and the revised manuscript. It profited from the revision. Unfortunately, I think it is still not yet publishable, especially regarding the justification and formulation of the hypothesis.

We wish to thank the reviewer for formulating their criticism in such a positive way.

  • The introductory sentences up until line 44 read vaguely and do not help to understand the topic. I suggest getting to the heart of the research question more quickly.

Thank you for this suggestion, we have now introduced the central issue from the beginning of the introduction

  • Many sentences lack proper referencing, especially:  ending at line 55 (several!), ending 58, 60, 64, 66, 81, 95 (!).

Thank you for pointing out missing information. We have now completed it.

  • revise sentence lines 77-79, it does not integrate and should be more informational. I think it is worth spending a sentence more on explaining the role of spontaneity 

We have now reformulated this and we hope it read more clearly.

  • Lines 100-109: The combination of "allowed us to demonstrate" and the formulation of a hypothesis does not fit. I am still not happy with the formulation of the hypothesis; it still does not align with the result section. Please revise and be more specific (H1: We expect that ... are less ...compared to..., H2: ...). 

Thank you again for seeking more clarity in our arguments. We have fully rewritten and expanded this section.

  • Theory summary: It is still not clear WHY you expect what you expect, and you are still too vague in formulating a precise prediction

We hope that reformulating and articulating more clearly our hypotheses (see previous point) also answer to this request.

  • Please be more scientifically neutral in your interpretation of lines 254-256. "Humbleness" reads a bit normatively influenced.

Thank you for pointing this issue out. It is certainly not our intention to follow this normative path… We have now chosen to refer to mindfulness.

  • Thank you for the opportunity to evaluate the revision!

Thank you for the opportunity to further improve our manuscript!

Round 3

Reviewer 1 Report

Comments and Suggestions for Authors

Thank you for the revision! 

The referencing and line of argument have substantially improved with your editing. The final point I would suggest is to choose a more careful formulation of the hypothesis: Instead of using "allowed us to demonstrate that" I would suggest stating it as a hypothesis/proposition (e.g., "therefore we expect that..."  (ll. 125-127)

Author Response

Comment 1: The final point I would suggest is to choose a more careful formulation of the hypothesis: Instead of using "allowed us to demonstrate that" I would suggest stating it as a hypothesis/proposition (e.g., "therefore we expect that..."  (ll. 125-127)

we have modified the last sentence of the introduction as proposed :

Therefore we investigated here whether Easterners exhibit a decreased tendency to consider an ego-centred perspective or a veridical and specific inclination to take other human’s perspective.

The new changes entailed to our ms are now highlighted in green.